# Machine Learning Methods for Diabetes Prevalence Classification in Saudi Arabia

Entissar S. Almutairi and Maysam F. Abbod *

Department of Electronics and Electrical Engineering, Brunel University London, London UB8 3PH, UK
* Correspondence: maysam.abbod@brunel.ac.uk

**Abstract:** Machine learning algorithms have been widely used in public health for predicting or diagnosing epidemiological chronic diseases, such as diabetes mellitus, which is classified as an epi-demic due to its high rates of global prevalence. Machine learning techniques are useful for the processes of description, prediction, and evaluation of various diseases, including diabetes. This study investigates the ability of different classification methods to classify diabetes prevalence rates and the predicted trends in the disease according to associated behavioural risk factors (smoking, obesity, and inactivity) in Saudi Arabia. Classification models for diabetes prevalence were developed using different machine learning algorithms, including linear discriminant (LD), support vector machine (SVM), K-nearest neighbour (KNN), and neural network pattern recognition (NPR). Four kernel functions of SVM and two types of KNN algorithms were used, namely linear SVM, Gaussian SVM, quadratic SVM, cubic SVM, fine KNN, and weighted KNN. The performance evaluation in terms of the accuracy of each developed model was determined, and the developed classifiers were compared using the Classification Learner App in MATLAB, according to prediction speed and training time. The experimental results on the predictive performance analysis of the classification models showed that weighted KNN performed well in the prediction of diabetes prevalence rate, with the highest average accuracy of 94.5% and less training time than the other classification methods, for both men and women datasets.

**Keywords:** machine learning; diabetes; classification





## 1. Introduction

Diabetes mellitus (DM) can be defined as "a group of metabolic diseases characterised by hyperglycaemia resulting from defects in insulin secretion, insulin action or both". DM-related disturbances in carbohydrate, fat, and protein metabolism are linked to chronic hyperglycaemia, and can result in long-term damage, dysfunction, and failure of various organs, especially the heart, eyes, kidneys, blood vessels, and nerves [1].

There are three types of DM classified according to aetiology and clinical picture: type 1 diabetes (T1DM), type 2 diabetes (T2DM), and gestational diabetes. T1DM is usually a result of absolute insulin deficiency due to the destruction of β cells in the pancreas, mostly due to a cellular-mediated autoimmune process. T2DM is caused by insulin resistance and relative insulin deficiency. Gestational diabetes is recognised or first starts during pregnancy, which is characterised by glucose intolerance of varying degrees of severity [2].

T2DM is the most common variant, accounting for 90% of diabetic diagnoses. People with T2DM are usually diagnosed after the age of 40, but younger adults or even children can be affected by this type. The symptoms of this type may not appear in the affected person for many years, and many patients are incidentally diagnosed due to seeking treatment for related or other complications. Unlike T1DM, patients with T2DM are not dependent on insulin therapy, but they may need insulin to control their hyperglycaemia if this is not reached with diet alone, or with oral hypoglycaemic agents [3].

T2DM is multifactorial, and its aetiology is complex. Different risk factors affect the incidence of the disease, but they are not all causative factors per se. These associated risk factors might be genetic, demographic (such as age), or factors related to the behaviour of the person, such as diet, smoking, obesity, and physical inactivity. Behavioural risk factors can be amended or changed; thus, they are often called "modifiable" risk factors [4]. T2DM is considered one of the most widespread noncommunicable diseases (NCDs) worldwide, with continually increasing prevalence. According to the International Diabetes Federation (IDF), there were more than 460 million people with diabetes in 2019, and it is expected that this figure will increase to 578 million in 2030 and 700 million in 2045. In the Kingdom of Saudi Arabia (KSA), the case of our study, there are currently an estimated 4 million diabetic patients according to the IDF [5].

During the past two decades, a variety of studies have sought to predict the incidence of diabetes and its global prevalence, using diverse data and methods of analysis [6,7]. Future estimates of the burden of diabetes are very important for health policy planning and identifying the necessary costs of controlling diabetes [8–10]. Recently, machine learning algorithms have been widely used in public health for predicting or diagnosing epidemiological chronic diseases, such as DM. Machine learning techniques can learn from the patterns of data and make predictions about unseen data. In machine learning, algorithms are used for data modelling, analysis, and visualization [11,12]. There are many published diabetes modelling studies using different machine learning techniques, including support vector machine (SVM), artificial neural network (ANN), k-nearest neighbour (KNN), and decision trees (DT) [13–16]. Most of these models are related to diagnosing and detecting diabetes at an early stage or modelling the disease progression and complications, but little has been carried out to adopt any of the machine learning classification methods in attempt to study the trends in the prevalence of diabetes and forecast its future burden using risk factors in specific populations such as in KSA; thus, this paper investigates the ability of different classification methods to classify diabetes prevalence rates and the predicted trends of the disease according to the related behavioural risk factors in KSA. The five classification groups of diabetes prevalence are labelled as: Class 1 (low), Class 2 (medium low), Class 3 (medium), Class 4 (high), and Class 5 (extremely high).

The paper is organized as follows. Section 2 reviews the literature. Section 3 presents modelling methodologies. Section 4 discusses experimental methodology. Section 5 presents the results. Section 6 provides a discussion of the obtained results. Finally, Section 7 concludes this paper and identifies areas for future research.

## 2. Literature Review

Machine learning algorithms have been widely used in public health for predicting or diagnosing epidemiological chronic diseases, such as DM. There are many published diabetes modelling studies which applying different machine learning techniques including SVM, ANN, KNN, and DT models or hybrid techniques. These models have been used for different purposes such as diagnosing or detecting diabetes at early stage, and for modelling the disease progression and complications. In this section, the commonly used machine learning algorithms are included with their accuracies in order to compare with the accuracies of the proposed classifiers. A summary of all the studies discussed in this section are provided in Table 1. In 2019, a comparative study conducted by Faruque et al. [17] used different machine learning models, including SVM, C4.5 decision tree, naïve Bayes, and KNN, and used the evaluation metrics of accuracy, recall, and precision to compare the performance of the classification models on predicting diabetes. In their study, they collected diabetes data from the diagnostic of Medical Centre Chittagong (MCC), Bangladesh. The dataset includes 200 patients with various attributes such as age, sex, weight, blood pressure, and other risk factors. The results obtained from this study indicated that the best performance was achieved by the C4.5 decision tree model with an accuracy of 73%. Another study in 2018 by Patil et al. [18] aimed to evaluate the performance of classification algorithms on the prediction of diabetes. In this study, the PIMA Indian data repository

was used, which included a total of 768 samples. These data were divided into training and testing sets with 70% for training (n = 583 samples) and 30% for testing (n = 230 samples). This study examined the implementation of eight machine learning models, namely: logistic regression (LR), KNN, SVM, gradient boost, decision tree, MLP, random forest, and Gaussian naïve Bayes. The results showed that the highest accuracy was achieved by the LR model, with 79.54% and an RMSE of 0.4652; the lowest accuracy was given by MLP, with 64.07% and an RMSE of 0.5994. The authors suggested improving the obtained results by using outlier detection before classification.

Additionally, in 2021, Bukhari et al. [19] constructed an ANN model using different numbers of neurons, from 5 to 50, in hidden network layers. In this study, the data were collected from the National Institute of Diabetes and Digestive and Kidney Diseases with the aim to predict female diabetes. Eight features were considered to train the ANN model, and the results showed an accuracy of 93% when using the Pima Indian Diabetes dataset for validation using the evaluation metrics of accuracy and mean squared error (MSE). In 2020, Hasan et al. [20] performed a study for diabetes diagnosis and prediction by using six machine learning models: AdaBoost, k-NN, decision trees, XG Boost, naïve Bayes, and random forest. The dataset used to train the models contained a total number of 768 female patients, with 268 diabetic patients (positive) and 500 nondiabetic patients (negative). In this study, eight different attributes were involved: glucose, insulin, pregnant, pressure, triceps, BMI, pedigree, pressure, and age. The procedures involved in preparing the data were feature selection, data standardization, outlier rejection, substitution with the mean for missing values, and k-fold cross-validation (5-fold cross-validation). An ensemble method was also implemented which was used to enhance the performance using multiple classifiers. In ensemble approaches, the combination of the outcomes obtained with different models can enhance the accuracy of the prediction. AdaBoost and XGBoost were the best models used together. For evaluating the performance, the area under the curve (AUC) was chosen as an evaluation metric. Their study was able to achieve an AUC score of 0.95 which was considered the best compared to other studies.

In 2021, Abdulhadi et al. [21] developed a variety of machine learning models for the purpose of predicting the presence of diabetes in females using the PIDD dataset. They addressed the problem of missing values using the mean substitution technique, and all attributes were rescaled using a standardization method. The constructed models were linear discriminant analysis (LDA), LR, SVM (linear and polynomial), and random forest (RF). Based on the results of their study, the highest accuracy score was achieved by the RF model, with 82%.

In another research work, in 2020, Oleiwi et al. [22] proposed a classification model aiming for the early detection of diabetes using machine learning algorithms. This study was designed to use significant features and deliver results which are close to the clinical outcomes. The data used in this study were collected from the patients using a direct questionnaire from the Diabetes Hospital of Sylhet, Bangladesh. This dataset includes reports of diabetes-related symptoms of 520 instances with 16 attributes. The authors used two class variables to find whether the patient had a risk of diabetes (positive) or not (negative). Three classification models were trained, namely a multilayer perceptron (MLP), a radial basis function network (RBF), and a random forest (RF), mainly to obtain the best classifier model for predicting diabetes. Their findings showed that the RBF model outperformed other models, with an accuracy of 98.80%.

Further to the studies that predicted or diagnosed diabetes, some existing studies have addressed the use of machine learning techniques to construct predictive models for diabetes complications. An example is the model developed in 2020 by Kantawong et al. [23] to predict some complications related to diabetes, particularly hyperlipidaemia, coronary heart disease, kidney disease, and eye disease. A dataset of 455 records was used in this study. Selection and cleaning processes were carried out on the dataset which reduced the number of records used to build the model. An iterative decision tree (ID3) algorithm was chosen to construct the model. For evaluating the performance of the proposed model,

a 10-fold cross validation method was used, giving an accuracy of 92.35%. It should be noted that the high score of accuracy obtained in this study is not sufficient to evaluate the performance of the model, specifically when training unbalanced data. The main reason for this is that when the model is trained, a minority class can be ignored, and all the predictions classified as the majority class still achieve good accuracy scores.

In 2018, Dagliati et al. [24] developed different classification models including LR, NB, SVMs, and random forest to predict the onset of retinopathy, neuropathy, and nephropathy in T2DM patients. The authors used different time scenarios for making the predictions: 3, 5, and 7 years from the first visit to the hospital for diabetes treatment. The dataset used to train the proposed models was collected by Istituto Clinico Scientifico Maugeri (ICSM), Hospital of Pavia, Italy for longer than 10 years. These data involve a total number of 943 records including the features of gender, age, BMI, time from diagnosis, hypertension, glycated haemoglobin (HbA1c), and smoking habit. The problem of unbalanced and missing data was managed by applying the missForest approach, while the problem of unbalanced classes was overcome by oversampling the minority class. The obtained results of this study show that the highest accuracy score was achieved by LR with 77.7%.

**Table 1.** Summary of machine learning classification techniques in diabetes research.

| Sr. No. | Researcher Name and Year | Methods/Techniques | Features | Limitations | Results and Findings |
|---|---|---|---|---|---|
| 1 | Faruque et al., 2019 [17] | SVM, C4.5 decision tree, naïve Bayes (NB), and KNN | -SVM works well with unstructured data. Additionally, it can provide good results even with less information about the data. -Solves different problems using an appropriate kernel function. | -Each variable has a different contribution to the output as the weights of these variables are not constant. -The training process takes a long time when using large datasets. | Researchers found that the C4.5 decision tree model outperformed the other classifiers with an accuracy of 73%. |
| 2 | Ratna Patil et al., 2018 [18] | Logistic regression (LR), KNN, SVM, gradient boost, decision tree, MLP, random forest, and Gaussian naïve Bayes | | | The highest accuracy was achieved by the logistic regression model, with 79.54% and an RMSE of 0.4652. |
| 3 | Bukhari et al., 2021 [19] | Artificial neural network (ANN) | -The ANN is easy to use and learns the given problem quickly. -Useful to solve problems where the input data are corrupted with noise. | -Learning can be slow. -It is difficult to know the required neurons and layers. | The ANN model showed an accuracy of 93%. |
| 4 | Hasan et al., 2020 [20] | AdaBoost, KNN, decision trees, XG Boost, naïve Bayes, and random forest | -KNN is easy to implement. -Suited for multiclass datasets. -Training is fast with good results and a high accuracy. | -It takes a long time to find the nearest neighbours in large datasets. -Memory limitation. | An ensemble method was implemented indicating that AdaBoost and XG Boost were the best models used together. This study achieved a score of area under the curve (AUC) of 0.95 which was considered the best compared to other studies. |

Table 1. *Cont.*

| Sr. No. | Researcher Name and Year | Methods/Techniques | Features | Limitations | Results and Findings |
|---|---|---|---|---|---|
| 5 | Abdulhadi et al., 2021 [21] | Linear discriminant analysis (LDA), LR, SVM (linear and polynomial), and random forest (RF) | -LDA easy to implement and fast in classification. -Can be applied to two or multiple classes. | -It is not appropriate for complex nonlinear data. -The distributions of LDA are non-Gaussian. -LDA fails if discriminatory information is based on the variance and not the mean of the data. | The highest accuracy score was achieved by the RF model, with 82%. |
| 6 | Oleiwi et al., 2020 [22] | Multilayer perceptron (MLP), a radial basis function network (RBF), and a random forest (RF) | | | Their findings showed that the RBF model outperformed other models, with an accuracy of 98.80%. |
| 7 | Kantawong et al., 2020 [23] | An iterative decision tree (ID3) algorithm | | | A 10-fold cross validation method was used, giving an accuracy of 92.35%. |
| 8 | Dagliati et al., 2018 [24] | LR, NB, SVM, and RF | | | The obtained results showed that the highest accuracy score was achieved by LR with 77.7%. |

## 3. Classification Methodologies

This section describes the selected machine learning classification methods: linear SVM, Gaussian SVM, quadratic SVM, cubic SVM, fine KNN, weighted KNN, linear discriminant (LD), and neural net pattern recognition (NPR). This is mainly to present the mathematical background of these methods and to highlight their individual operational characteristics.

### 3.1. Support Vector Machine

SVM is one of the most commonly used supervised machine learning algorithms, which can be used for both regression analysis and classification tasks. SVMs are considered as new types of pattern classifiers related to learning algorithms, which can identify patterns and analyse data. They have been successfully applied in various types of applications such as verification and detection, text categorisation and detection, recognition information, recognition of handwritten characters, speakers, and speech verification [25]. SVM can perform linear classification and predict nonlinear separable patterns by mapping its inputs into a hyperplane (high-dimensional feature spaces). Minimising the upper bound of the generalisation error is the main aim of SVM by maximising the transaction between the data and the separating hyper plane. The performance of SVMs tends to be accentuated when using new data not included through the training process because of its fundamental classification principle, which produces new examples into the related class. In a standard classification case, the components of the dataset include several parameters X1, X2, ... , X3 and one or multiple variables for classes C1, C2, ... , CP. The objective is to develop a classifier to appoint the inputs (data points) to their classes (C1, C2, ... , CP) by using the N data points in the training set. Thus, for every point in the training set $\{x_n\}_{n=1}^{N}$, a class $t_n$ should be predicted where $t_n \in \{-1, 1\}$, n = 1, ... ,N. Then, the classifier can be defined in Equation (1):

$$y(x, w) = \sum_{i=1}^{J} w_i \varphi_i(x) + b \tag{1}$$

where $w \in R^J$ is the weight vector; $\varphi(.)$ is the transformation function; and b $\in$R is the constant. If the data space is nonlinearly separable, SVMs use an appropriate mapping ($\varphi$) of the input data to a high-dimensional space which will be arranged by the kernel function. The kernel function is defined as follows:

$$K(x, x') = \varphi(x).\varphi(x') \tag{2}$$

Separating the hyperplane by a kernel solution does not require knowledge of $\varphi$, but only of K. Therefore, any kernel function can be used in a larger dimension space. Different kernel functions can be most efficient; this depends on the nature of the dataset. The following sections describe the used kernel functions.

### 3.1.1. Linear SVM

The linear kernel function is one of the simplest types of kernel functions and the basic way to use SVM classifier. It is calculated by the inner product of two vectors, $x_i$ and $x_j$, plus an optional constant, $b$. Thus, the kernel algorithms using a linear kernel are often the same as their non-kernel equivalents. The linear kernel provides a signed measure of the similarity between $x_i$ and $x_j$, where the angle between the two points helps determine their similarity and can give negative values of $K$. The following equation shows the linear kernel function:

$$K\langle x_i, x_j \rangle = x_i^T.x_j + b, \tag{3}$$

where $b$ is a constant.

### 3.1.2. Gaussian SVM

The Gaussian SVM kernel function is a type of radial basis function (RBF), which represents the most generalised form of kernelization and the most commonly used kernels. It can be described as a general-purpose kernel, and it is used when there is no previous information regarding the data. Unlike the linear kernel, the Gaussian kernel only relies on the Euclidean distance between $x_i$ and $x_j$, and it also depends on the assumption that similar points in the feature space are close to each other [26]. This final assumption is logical in many situations; thus, the Gaussian kernel is widely used in practice. The Gaussian kernel can be mathematically represented as follows:

$$K(x_i, x_j) = e^{-\gamma \|x_i - x_j\|^2} \tag{4}$$

where $\gamma > 0$ is a given parameter, which must be accurately tuned as it directly affects the kernel performance.

### 3.1.3. Quadratic SVM

Quadratic kernel is a common form of polynomial kernel, commonly applied in speech recognition. The computation of this function is less intensive than the Gaussian kernel function and can be alternatively used if using the Gaussian kernel is too expensive. These functions do not generalize well because higher-order kernels tend to overfit the training data. The quadratic function can be mathematically expressed as follows:

$$K(x_i, x_j) = 1 - \frac{\|x_i - x_j\|^2}{\|x_i - x_j\|^2 + b} \tag{5}$$

3.1.4. Cubic SVM

The SVM classification method is helpful when facing a problem of low memory space. SVM can find a hyperplane in multidimensional space which separates the label classes in the best possible way. The cubic SVM classifier is a form where the kernel function of the classifier is cubic, and its mathematical expression can be given as the following:

$$K(x_i, x_j) = (x_i^T x_j + 1)^3 \qquad (6)$$

*3.2. K-Nearest Neighbours*

KNN is one of the simplest machine learning methods that can be used for regression as well as for classification, but it is commonly used for the classification problems in classifying data input into predefined classes (k). This method dates back to 1968 when it was first introduced by Cover and Hart [27]. The KNN method is considered as a nonparametric method; this means no assumptions will be made on the involved data. Additionally, this algorithm does not immediately learn from the training data, but rather it stores the dataset and, at the time of classification, it makes an action on the dataset, so it is called a lazy learner algorithm. In this method, the input data include all the closest training points in the feature space, where k is an integer.

The data are classified by determining the most common class among the k-nearest neighbours. These neighbours are members in the dataset in which this method was first trained and are identified using the distance from the test sample. This means that during the testing, the class which appears most commonly amongst the neighbouring classes of the test sample under observation becomes the class to which this individual test sample belongs. In other words, after the training phase, any new data obtained are classified into a similar category to these new data. The accuracy of the KNN classifiers is increased with a decreasing number of neighbours. This leads to increasing the complexity of the classifier model; however, it does not ensure that the new samples will be classified correctly. There are many types of distance function between the samples; the most popular used function is the Euclidean distance, which is presented in the following equation:

$$d = \sqrt{\sum_{k=i}^{n} (X_{ik} - X_{jk})^2} \qquad (7)$$

where $X_1$ and $X_2$ are input samples, and *k* is the number of values in each sample.

There are six types of KNN classifiers available in MATLAB: fine KNN, medium KNN, coarse KNN, cosine KNN, cubic KNN, and weighted KNN. Some of these types of KNN algorithms make use of Euclidean distance to determine the nearest neighbours (fine, medium, and coarse KNN algorithms). The cosine KNN algorithm employs a cosine distance metric as given in Equation (8). For the cubic KNN algorithm, a cubic-distance metric is employed as in Equation (9). For the weighted KNN algorithm, a distance weight is employed, as in Equation (10).

$$d = \left( 1 - \frac{x_i \, x_j'}{\sqrt{(x_i \, x_i')(x_j \, x_j')}} \right) \qquad (8)$$

$$d = \sqrt[3]{\sum_{k=i}^{n} |x_{ik} - x_{jk}|^3} \qquad (9)$$

$$d = \sqrt{\sum_{k=i}^{n} w_i (X_{ik} - X_{jk})^2} \qquad (10)$$

Only fine KNN and weighted KNN were chosen along with other classifiers to classify our data.

### 3.3. Linear Discriminant Analysis

Linear discriminant analysis (LDA), also known as Fisher's LDA, is a commonly used statistical machine learning method for data classification and dimensionality reduction problems [28]. The main aim of LDA is to find the best linear approximations of object feature vectors for efficient and sensible use in a variety of classification tasks. The idea behind this method is maximising the ratio of between-class variance to the within-class variance in any specific dataset, in that way ensuring maximal separability. In other words, the aim is to utilise a linear transformation process for projecting high-dimensional feature vectors into a lower dimension to best separate the data groups [29]. The advantage of LDA is its implementation simplicity, whereby a linear combination of features is used for distinct classes of samples. Furthermore, it easily manages cases where within-class frequencies are unequal, and their performance has been assessed on randomly generated test data. On the other hand, the simplicity of its execution leads to a drawback, particularly if the class differences are low. In this case, LDA assumes the mean as a discriminating factor and not the variance, which in turn leads to overfitting the data.

### 3.4. Neural Network Pattern Recognition

Pattern recognition is one of the most important aspects of computer science, which can be defined as the process of observing a system or event to recognise and distinguish patterns by utilising machine learning methods. It could also be defined as a data-analysis approach for classifying the data based on their background or statistical information derived from patterns, and then making reasonable decisions about the categories of the patterns. Its advantages include its ease of use and the practical possibility of widespread application. Despite many years of research, the design of a multipurpose machine for pattern recognition is still a long way off. According to Ross [30], "The more relevant patterns at your disposal, the better your decisions will be". This is promising news for AI supporters, as it is possible to teach computers how to recognize patterns. In fact, a variety of successful computer programs have been used in diagnosing diseases, bank credit scoring, and landing airplanes which, in some way, depend on pattern recognition. There are two possible ways which can be used in recognising patterns including classification and cluster. Classification is a supervised-learning approach where a proper class is appointed according to a pattern that is extracted from training datasets, whereas clustering is unsupervised learning, and it works by dividing the data into groups which in turn help in making decisions. In this paper, the artificial neural network method was used for pattern recognition as a classification method.

ANNs can be defined as statistical models with a nonlinear way of modelling the data which simulate the way of biological NNs. Statistical-pattern-based methods have been the most common and applied in different practices. However, ANN models have gained more popularity and attractiveness due to offering more efficient and successful ways of dealing with pattern-recognition problems in many cases.

In contrast with conventional pattern methods, ANN can easily handle complex or multicomplicated tasks. The conventional methods which are used to deal with pattern-recognition problems can be divided into three types: statistical, structural, and hybrid methods. However, insufficient results can be obtained using both the statistical and structural methods when they are used to solve the complicated pattern-recognition problems only. For example, when applying the structural method, the performance can be poor and unable to handle noise patterns. In the same way, the statistical method is incompetent in dealing with information related to pattern tasks. Therefore, the combination of the two methods was widely accepted by researchers which in turn led to the hybrid method. However, at present, ANN models can be used instead of the conventional hybrid methods; this is because of the good results obtained in pattern recognition even in more complicated problems [31].

## 4. Experimental Methodology

This study used historical data on the prevalence of diabetes, smoking, obesity, and inactivity in Saudi Arabia from 1999 to 2013, to achieve the study aim and develop the models. The main sources of data were the published national surveys in KSA. Data on the prevalence of diabetes, smoking, obesity, and inactivity in KSA were obtained from a Saudi Health Interview Survey [32], which was provided by the Saudi Ministry of Health, along with other published national surveys [33–36].

All these population-based studies were implemented at the national level, included all regions in KSA, and used good sampling techniques of multistage stratified random sampling to recruit the study subjects of both sexes with response rates ranging from 90 to 97%. Thus, they were more likely represent the population of KSA. These population-based national studies include adults aged 15 years and over. In addition, the diagnostic criteria used as a diabetes-detection method were either World Health Organisation (WHO) or American Diabetes Association (ADA) criteria. In this study, obesity as a risk factor was defined according to the definition of body mass index (BMI $\geq$ 30 kg/m$^2$); for smoking, only data for current smokers were taken; and for inactivity, inactive people were classified as those who did not meet the criteria for the "active" category (30 min or more of at least moderate-intensity activity three or more times per week). The dataset has 5 attributes (age, gender, smoking, obesity, and inactivity) and 1272 entries (men and women aged 25 and above). A total of 840 entries were used for the training stage, and 432 were used for testing. The data were divided into six ten-year age bands (25–34, 35–44, ... 75+ years old) for men and women. These data were recently used in our previous study to develop regression methods to predict the prevalence rate of diabetes up to 2025. In that study, we showed that there was a steady increase in diabetes prevalence among men and women in all age groups, but the prevalence was lower in younger age groups, and it increased with age. The findings showed that the highest prevalence of diabetes from all the age groups was found in the population aged from 55 to 74 years. Additionally, a correlation analysis was carried out to determine the correlation between the variables in the data in terms of statistical significance. The findings indicated that demographic (age and gender) as well as behavioural risk factors significantly contributed to the increased level of diabetes, with a significance level of 0.05; however, smoking, obesity, and physical inactivity were the most significant factors. Details on the developed regression models can be found in [37].

In this paper, the implementation of the proposed methodology was carried out using a Dell PC with Intel$^®$ Pentium$^®$ CPU N3710, 4 GB o memory and 1.6 GHz processor. Several classification methods were employed to classify the prevalence rate of diabetes disease into five different classes using different classification methods: LD, SVM with linear kernel, SVM with Gaussian kernel, SVM with quadratic kernel, SVM with cubic kernel, fine KNN, weighted KNN, and NPR. All these methods were implemented in MATLAB (version R2018a) using the Classification Learner App, except the neural network pattern recognition method, which was implemented using the Neural Net Pattern Recognition App [38].

Data were prepared for classification by converting the continuous values of diabetes prevalence into discrete using the (discretize) function in MATLAB based on five classes for both men and women datasets: Class 1, low (7% to 10.5%); Class 2, medium low (8.55% to 10.9%); Class 3, medium (10.71% to 14.8%); Class 4, high (13.02% to 16.15%); and Class 5, extremely high (15.61% to 17.59%). All the datasets of the three risk factors (inputs) with the labelled classes of diabetes morbidity data (outputs) were fitted into the (classification learner) for the training stage, using the default option of 5-fold cross-validation for each method. Figure 1 demonstrates the training process of the classification models.

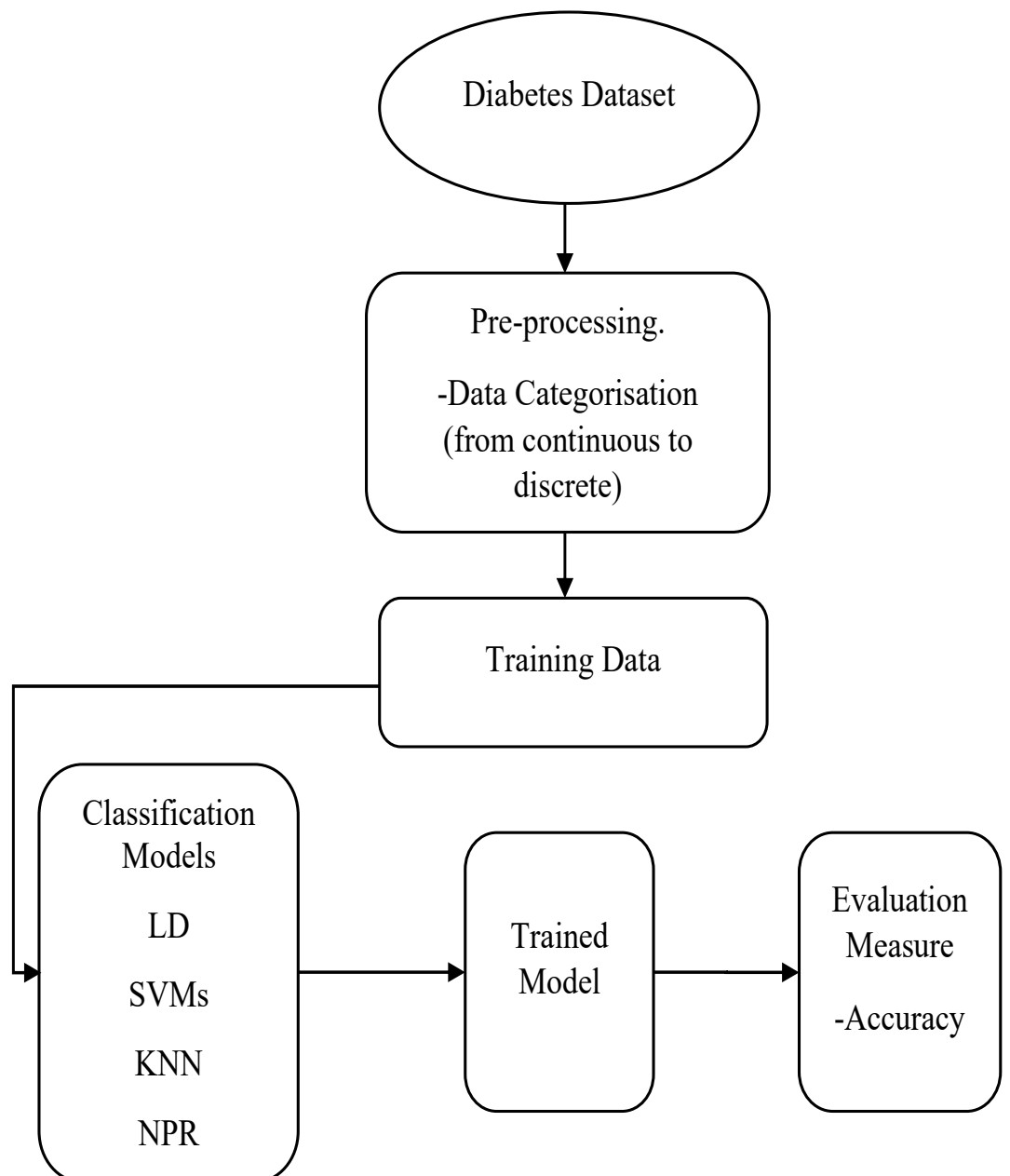

**Figure 1.** Flowchart of the training process of the classification models.

For the SVM classification method, the characteristics of this algorithm regarding the kernel function, kernel scale, and other parameters are given in Table 2. One of these parameters is the (box constraint level) which is a parameter that controls the maximum penalty imposed on margin-violating observations and helps in avoiding overfitting. If the box constraint level is increased, then fewer support vectors will be assigned; however, increasing the box constraint can result in longer training times.

**Table 2.** Characteristics of SVM classification models.

| Classifier Characteristics | Model 1 | Model 2 | Model 3 | Model 4 |
|---|---|---|---|---|
| Preset | Linear SVM | Quadratic SVM | Cubic SVM | Medium Gaussian SVM |
| Kernel function | Linear | Quadratic | Cubic | Gaussian |
| Kernel scale | Automatic | Automatic | Automatic | 1.7 |
| Box constraint level | 1 | 1 | 1 | 1 |
| Multiclass method | One-vs-One | One-vs-One | One-vs-One | One-vs-One |
| Standardize data | true | true | true | true |

In addition, the information of the KNN classification method, such as the number of neighbours, distance metric, and distance weight, are shown in Table 3. After that, the pretrained classifiers were exported to the MATLAB workspace where they were used to make predictions for data chosen randomly from our data points using the MATLAB function predictFcn. This testing stage tested the accuracy of the classification models.

**Table 3.** Characteristics of KNN classification models.

| Classifier Characteristics | Model 1 | Model 2 |
|---|---|---|
| Preset | Fine KNN | Weighted KNN |
| Number of neighbours | 1 | 10 |
| Distance metric | Euclidean | Euclidean |
| Distance weight | Equal | Squared inverse |
| Standardize data | true | true |

For the pattern recognition classification method, the Neural Net Pattern Recognition app in MATLAB was used, which can be used directly from the Apps tab or by typing (*nprtool*) in the command window. In this app, a two-layer feed-forward network with a sigmoid hidden layer and softmax output neurons were used, and the network was trained with scaled conjugate gradient backpropagation. The structure of the network consisted of input, hidden, and output layers. Each layer had a number of neurons or elements; in our experiment, 3 neurons were used in the input layer (representing the risk factors), and 5 neurons in the output layer (representing the five classes). There is no exact way to define the number of neurons in the hidden layer, and it is commonly chosen according to the trial-and-error method to obtain the best network performance, which in this experiment was achieved with 10 neurons. The performance of all classifiers was evaluated by accuracy. In addition, for the classification learner's methods, further comparisons were conducted between the classifiers according to the prediction speed and training time. All the results are presented and discussed in the next sections.

## 5. Results

This section presents the obtained results from each classification model discussed in the previous section. The performance evaluation in terms of the accuracy of each developed model was determined, and the developed classifiers were compared using the Classification Learner App according to prediction speed and training time.

All classification methods were compared according to their accuracy as shown in Tables 4 and 5 for the men and women datasets, respectively. For the men data in Table 4, Gaussian SVM, weighted KNN, and neural network pattern recognition models gave the same accuracy (92.6%), which was the highest value achieved among the models. The remaining models also performed well, with a similar accuracy of 88.9%. Table 5 provides different accuracies given by models when applied to the women data. The highest accuracy was given by Gaussian SVM, fine KNN, and weighted KNN models of 96.3%, followed by neural network pattern recognition with an accuracy of 92.6%. Additionally, three other models had good performance with the same accuracy of 85.2%: linear discriminant, quadratic SVM, and cubic SVM. Lastly, the linear-SVM-based model

had the lowest accuracy of 77.8%. In addition, Tables 4 and 5 also show a comparison of classifiers in terms of prediction speed (the number of observations processed per second during the validation of the model) and training time. It is clearly shown from Tables 4 and 5 that weighted KNN model was the fastest with the highest prediction speed of 310 obs/s and 340 obs/s for the men and women datasets, respectively. On the contrary, linear SVM was the slowest with 77 obs/s and 110 obs/s for the men and women datasets, respectively. The maximum training time was taken by linear SVM with 10.177 s and 5.4628 s, while the weighted KNN took the least training time of 1.2459 s and 1.6991 s for the men and women datasets, respectively. Overall, among all classifiers, the weighted KNN model performed well in terms of accuracy, prediction speed, and training time for both the men and women datasets.

**Table 4.** Classification outcome information (men data).

| Classifiers | Accuracy | Prediction Speed | Training Time |
|:---:|:---:|:---:|:---:|
| LD | 88.9% | 93 obs/s | 9.9083 s |
| Linear SVM | 88.9% | 77 obs/s | 10.177 s |
| Quadratic SVM | 88.9% | 100 obs/s | 4.3084 s |
| Cubic SVM | 88.9% | 120 obs/s | 3.3928 s |
| Gaussian SVM | **92.6%** | 120 obs/s | 3.7141 s |
| Fine KNN | 88.9% | 140 obs/s | 3.7141 s |
| Weighted KNN | **92.6%** | 310 obs/s | 1.2459 s |
| NPR | **92.6%** | … … … . . | … … … … . . . |

**Table 5.** Classification outcome information (women data).

| Classifiers | Accuracy | Prediction Speed | Training Time |
|:---:|:---:|:---:|:---:|
| LD | 85.2% | 310 obs/s | 2.2019 s |
| Linear SVM | 77.8% | 110 obs/s | 5.4628 s |
| Quadratic SVM | 85.2% | 91 obs/s | 4.6329 s |
| Cubic SVM | 85.2% | 100 obs/s | 4.2164 s |
| Gaussian SVM | **96.3%** | 110 obs/s | 3.5432 s |
| Fine KNN | **96.3%** | 320 obs/s | 1.5032 s |
| Weighted KNN | **96.3%** | 340 obs/s | 1.6991 s |
| NPR | 92.6% | … … … . . | … … … … . . . |

In addition, a graphical representation of the confusion matrixes of the classification models is clearly displayed in Figures 2–8 for both men and women datasets, respectively. For each model, the diagonal green squares represent the correct prediction ratio of the predicted classes over true classes, while the red squares give the incorrect class ratio. The results show the level of the decision ratio between the true value and the predicted value. Figure 2 shows the confusion matrix of the predicted and true class categories obtained by the LD model; for the men data, 0.09 of the data were misclassified for the following classes (class 1, class 2, and class 3). For the women data, 0.12 of the data were misclassified for class 2 and class 3. So, the overall classification accuracy of the LD model was 88.9% and 85.2% for the men and women datasets, respectively. Figure 3 shows the maximum misclassification among models obtained by LSVM; 0.09 of the data were misclassified for both men and women data for the classes 1, 2, 4, and 5, which gave an average classification accuracy of 83.4% for the men and women datasets. Figures 4 and 5 demonstrate the confusion matrixes of the predicted and true class obtained by quadratic SVM and cubic SVM for men and women, respectively; as can be seen, 0.21 of the data were misclassified for both models for men and women data for the classes 1, 2, 4, and 5, with a classification accuracy of 88.9% for both models for men and 85.2% for both

models for women, respectively. Figures 6–8 present the confusion matrixes of the best performing classifiers, Gaussian SVM, fine KNN, and weighted KNN; the misclassification of class ranged between 0.03 and 0.11 for all models for men and women, with an average classification accuracy of 93.8% for men and women datasets.

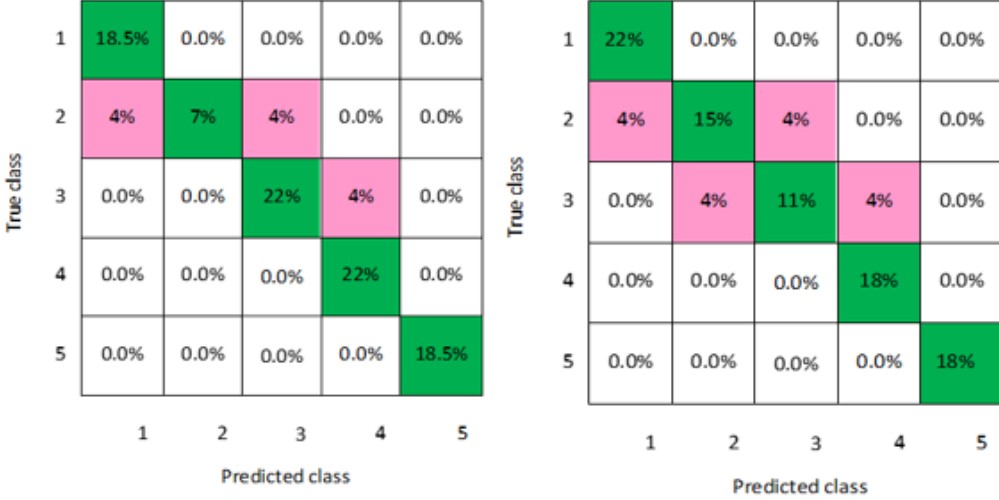

**Figure 2.** LD model confusion matrixes for men and women data, respectively.

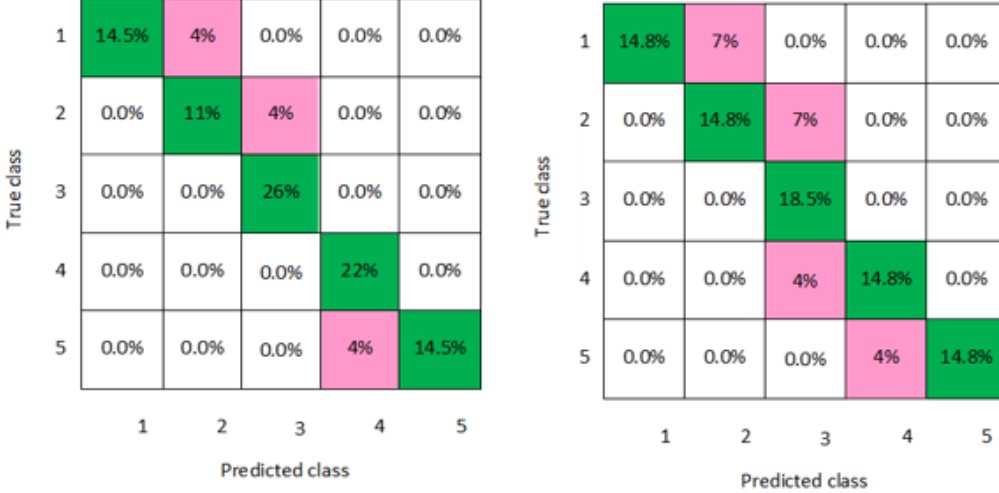

**Figure 3.** Linear SVM model confusion matrixes for men and women data, respectively.

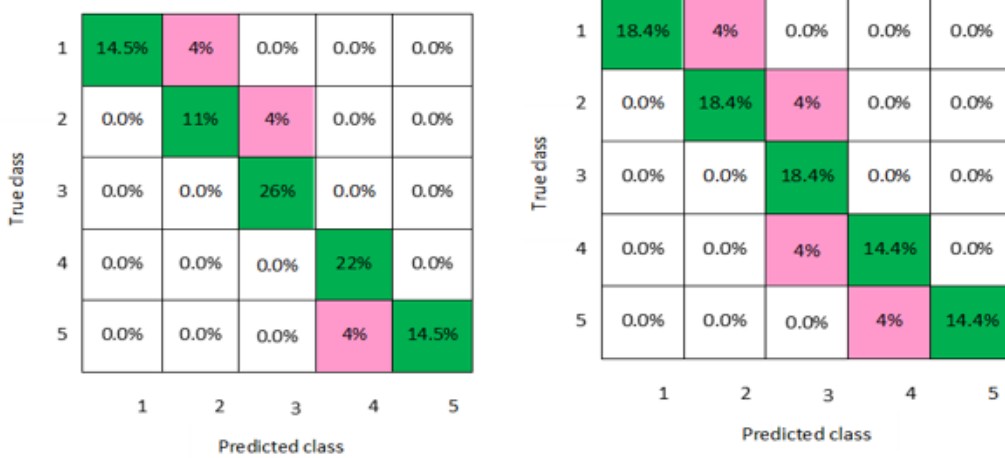

**Figure 4.** Quadratic SVM model confusion matrixes for men and women data, respectively.

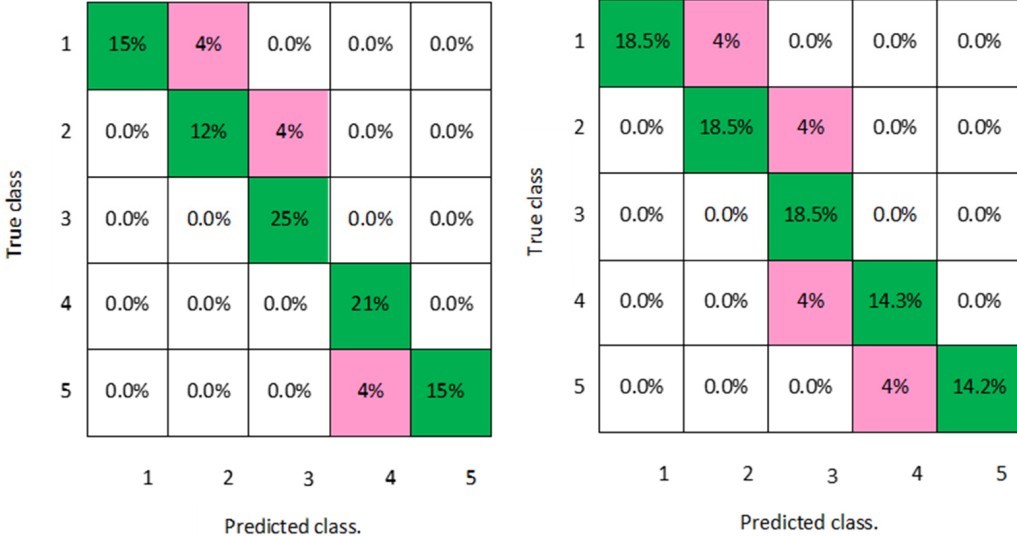

**Figure 5.** Cubic SVM model confusion matrixes for men and women data, respectively.

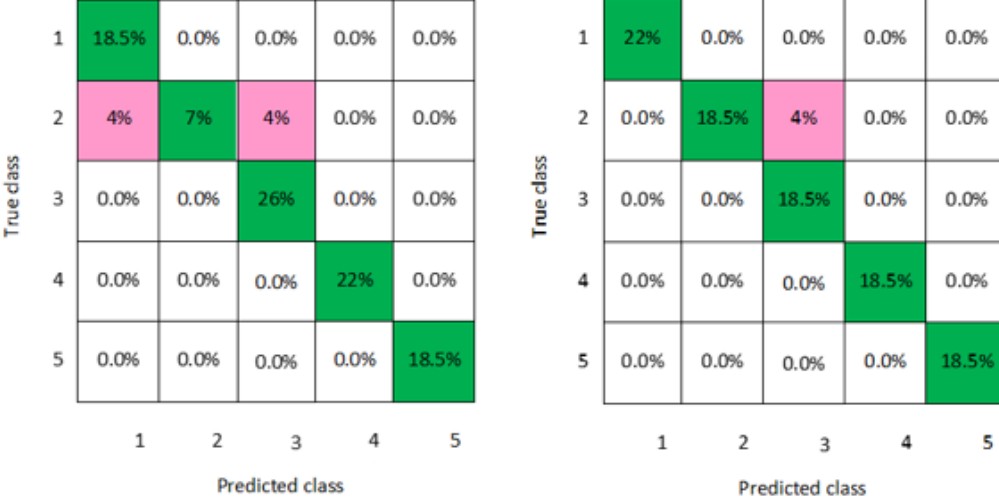

**Figure 6.** Medium Gaussian SVM confusion matrixes for men and women data, respectively.

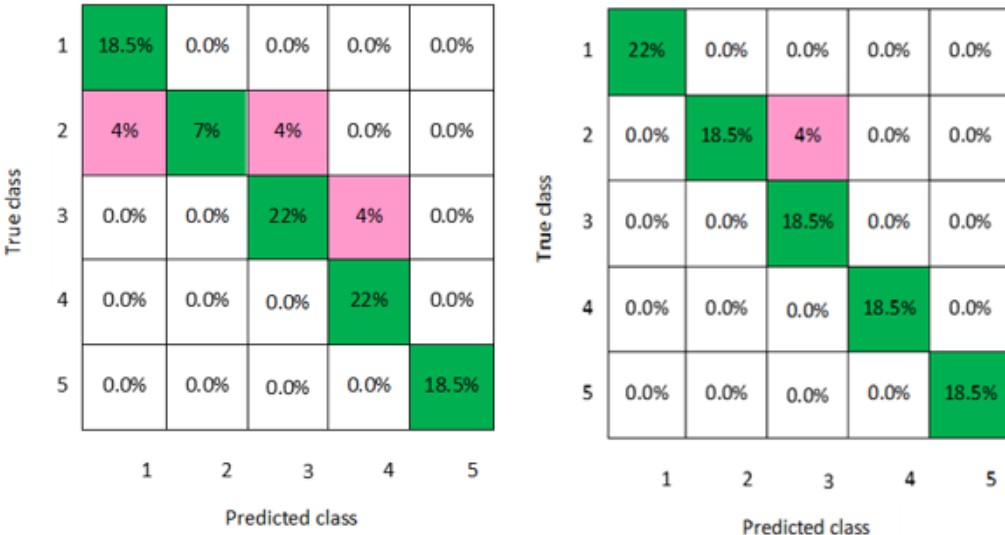

**Figure 7.** Fine KNN confusion matrixes for men and women data, respectively.

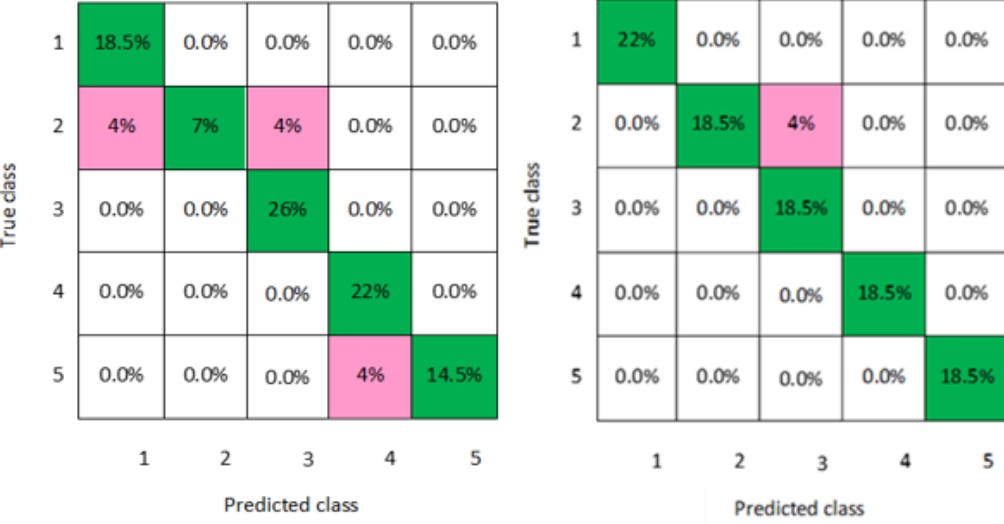

**Figure 8.** Weighted KNN confusion matrixes for men and women data, respectively.

## 6. Discussion

The obtained results illustrated by the confusion matrixes of the developed models using men datasets show that the error rate for diabetes prevalence classification was the same for all models (3%) except for Gaussian SVM (2%). For the women data, the achieved results show that the same error rate was given by LD, quadratic SVM, and cubic SVM (4%), while linear SVM gave a maximum error rate of 6%. Moreover, the best results again were achieved by the Gaussian SVM, fine KNN, and weighted KNN models, all indicating the same error rate of 1%. Figure 9 compares the classification results of all models in terms of accuracy for both the men and women datasets. It can be seen that the performance of the Gaussian SVM, fine KNN, and weighted KNN models was better for the women data than for the men data, while the performance accuracy of LD, linear SVM, quadratic SVM, and cubic SVM was higher when using men data than women data. The differences between the men and women datasets affected the performance of the models, particularly with SVM models, as indicated in Table 1. For SVM, there is a disadvantage that different parameters have a different influence on the prediction accuracy as the weights of these variables are not constant. The NPR model had the same performance accuracy for both men and women datasets.

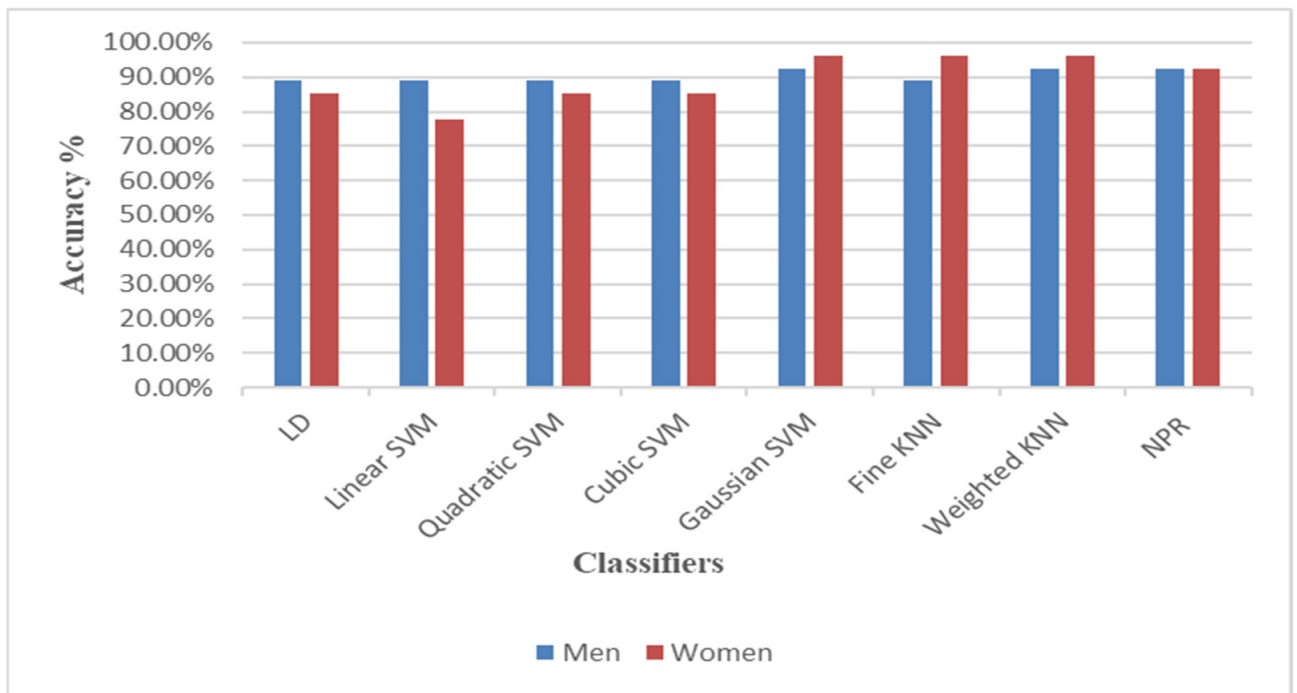

**Figure 9.** Classification results (accuracy) for men and women datasets.

The accuracies of the proposed classifiers (average SVM with different kernel functions, fine KNN, and weighted KNN) were compared to the same type of classifiers in other studies as shown in Table 6. The comparison demonstrated that the accuracies of the proposed classifiers were higher than those obtained with these classification techniques in previous studies. In the present study, the SVM classification method was used with different functions in order to achieve the best results with high accuracy. In comparison to previous studies, they only applied a linear SVM which might not be an effective kernel function for their datasets. The average accuracy of all SVMs with different kernels and KNN models ranged between 87.9% and 93.6% for the men and women datasets. However, good performance was achieved by the weighted KNN model in terms of accuracy, prediction speed, and training time for both men and women datasets. This means that the proposed classifiers perform successfully in diabetes prediction.

**Table 6.** Comparison of the accuracy of SVM and KNN classifiers with other studies.

| Sr. No. | Researcher Name and Year | Classification Accuracy |
|---------|--------------------------|-------------------------|
| 1 | Faruque et al., 2019 [17] | SVM 79%<br>KNN 70% |
| 2 | Ratna Patil et al., 2018 [18] | SVM 68%<br>KNN 75% |
| 3 | The proposed SVM and KNN Classifiers | SVM **87.9%**<br>Weighted KNN **94.5%**<br>Fine KNN **92.6%** |

## 7. Conclusions and Future Work

This paper investigated the use of classification methods to group or classify the prevalence level of diabetes disease that was associated with behavioural risk factors (smoking, obesity, and inactivity). For this purpose, different classification machine learning models have been developed based on Linear discriminant, Linear SVM, Quadratic SVM, Cubic SVM, Gaussian SVM, Fine KNN, Weighted KNN, and Neural Net pattern recognition. The proposed methods were applied on the men and women datasets; first, the morbidity data of diabetes were prepared for classification by converting the continuous values of diabetes dataset into discrete. Second, all the datasets of the three risk factors (inputs) with the labelled classes of diabetes morbidity data (outputs) were fitted into the (classification learner) for the training stage. Finally, the performance of these models was evaluated in terms of accuracy, and they were further compared according to the training time and prediction speed. The obtained results show that there were slight differences in the performance of models when using the men and women datasets. In addition, the experimental results on the predictive performance analysis of the classification models showed that Weighted KNN performed well in the prediction of diabetes prevalence rate compared to the other classification methods, with the highest average accuracy of 94.5 and less training time for both men and women datasets. It was the fastest with the highest prediction speed of 310 obs/s and 340 obs/s, and it took the least training time of 1.2459 s and 1.6991 s for men and women datasets, respectively. The findings indicate that the developed models can accurately classify the prevalence of diabetes into different classes.

In the future, it could be useful to investigate the effect of including more risk factors for diabetes in KSA, such as diet or blood pressure, or even including different categories of risk factors (e.g., nonmodifiable risk factors), such as family history and gestational diabetes. The integration of different risk factors might help to obtain more precise predictions of diabetes prevalence. Additionally, examining the use of machine learning methods to predict the risk of developing any complications of diabetes such as nephropathy, retinopathy, and cardiovascular disease could also be useful. This would help to maintain the quality of life of diabetic individuals and reduce the rising burden of diabetes on healthcare budget.

**Author Contributions:** M.F.A. and E.S.A. developed the algorithms, analysed the data, and wrote the paper. M.F.A. evaluated and supervised this study. All authors have read and agreed to the published version of the manuscript.

**Funding:** This research received no external funding.

**Institutional Review Board Statement:** Not applicable.

**Informed Consent Statement:** Not applicable.

**Data Availability Statement:** Data presented in the paper are available on request from the corresponding author M.F.A.

**Conflicts of Interest:** The authors declare no conflict of interest.

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
