# Peer review of "Machine Learning Methods for Diabetes Prevalence Classification in Saudi Arabia"

_2673-3951, doi:10.3390/modelling4010004_

Round 1

Reviewer 1 Report

In this study, the authors compared machine learning methods for diabetes prevalence classification.

My comments are listed below:

1. Why did the authors divide the dataset into "men" and "women"? No information on this was provided.

2. The authors examined the relationship between risk factors and classification (line 414 and line 525). They did not address whether these factors were related to age and gender. Why and how this path is approach to obtain a more generalized model is not explored.

3. In this study, the authors used data from 1999 to 2013. Is there any data available after 2013? If more recent data is available, why was your data set chosen in this way?

4. There are similar studies on this subject in the literature. The authors do not sufficiently highlight the innovation that this study has.

5. It is seen that the study was carried out on MATLAB. Innovation in the study was limited by the authors' use of ready-made tools and models.

6. The Discussion section is quite inadequate. Only the results are written in sentences, but the reasons for the results are not examined in detail. In addition, the results were not adequately compared.

7. What does the "box constraint level" parameter in Table 2 mean? If it is a parameter related to SVM, its description should be given.

8. Conclusion section should be extended with numerical results.

9. The quality of the figures should be improved.

10. Figures 2-8 are screenshots taken directly from the MATLAB app screen. It is recommended that these confusion matrix plots be drawn in academic style to make the paper look more academic.

Author Response

Reviewer 1

  1. Why did the authors divide the dataset into "men" and "women"? No information on this was provided.

Response 1: this paper is part of a research which we also used regression modelling techniques to study the trends in the prevalence of diabetes and forecast its future using risk factors in specific populations such as in KSA.

One of the main objectives was to Identify the available local data which are required for developing the models. The required data include prevalence of diabetes, smoking, obesity, and inactivity, also the demographic data (age and gender) in KSA. (Dataset was divided into "men" and "women because the differences between them), for example smoking prevalence is higher among men than women.

  1. The authors examined the relationship between risk factors and classification (line 414 and line 525). They did not address whether these factors were related to age and gender. Why and how this path is approach to obtain a more generalized model is not explored.

Response 2: more explanation has been added in lines 372 to 382.

  1. In this study, the authors used data from 1999 to 2013. Is there any data available after 2013? If more recent data is available, why was your data set chosen in this way?

Response 3: As mentioned before that this study is a part of a research where the data were extracted according to 6 age groups for (men and women) and for each behavioural risk factor (smoking, obesity, and inactivity). Some years between 1999 to 2013 were available and some were not, for the missing years we used linear interpolation imputation technique to estimate these data assuming a linear increase in the prevalence of diabetes and the risk factors. Some of the available data after 2013 were only for the overall prevalence of diabetes.

  1. There are similar studies on this subject in the literature. The authors do not sufficiently highlight the innovation that this study has.

Response 4: This point has been highlighted in lines 65 to 71, … most of these models have been related to diagnosing and detecting diabetes at early stage or modelling the disease progression and complications, but little has been done to adopt any of the machine learning classification methods in attempt to study the trends in the prevalence of diabetes and forecast its future burden using risk factors in specific populations such as in KSA, thus this paper investigates the ability of different classification methods to classify diabetes prevalence rates and the predicted trends of the disease according to the related behavioural risk factors in KSA.

The aim of our study is studying the trends in the prevalence of diabetes (the epidemiology of the disease) and forecast its future level using risk factors in specific populations such as in KSA.

  1. It is seen that the study was carried out on MATLAB. Innovation in the study was limited by the authors' use of ready-made tools and models.

Response 5: As mentioned before that this study is a part of a research where different regression models and combination methods were used, this needs programs to be written such as the consensus and fuzzy combiners. For the classification modelling in this paper, classification techniques were used which were provided by MATLAB to help for modelling and simulation.

  1. The Discussion section is quite inadequate. Only the results are written in sentences, but the reasons for the results are not examined in detail. In addition, the results were not adequately compared.

Response 6: some improvements have been added in the results and discussion section.

  1. What does the "box constraint level" parameter in Table 2 mean? If it is a parameter related to SVM, its description should be given.

Response 7: The "box constraint level" is a parameter related to SVM classifier. (Description about it has been included in lines 406 to 410.

  1. Conclusion section should be extended with numerical results.

Response 8: Numerical results have been added.

  1. The quality of the figures should be improved.

Response 9: Figures have been improved.

  1. Figures 2-8 are screenshots taken directly from the MATLAB app screen. It is recommended that these confusion matrix plots be drawn in academic style to make the paper look more academic.

Response 10: Figures have been improved.

Reviewer 2 Report

In this manuscript, Almutairi and Abbod reported the Machine learning methods for diabetes prevalence classification in Saudi Arabia. A thorough reading of manuscript indicates that it fits the scope of journal. This manuscript can be considered for publications after minor revision; I have following points:

(1) Purpose of study need to clearly explain.

(2) Literature Review is unnecessarily lengthy.

(3) More information is required about data.

(4) Conclusion and Future Work, detail about Future Work is missing.

(5) Need to cite the literature about machine learning such as

10.1007/s10118-022-2782-5, 10.1039/D1TA04742F , 10.1002/chem.202103712 , 10.1039/d1ta09762h

Author Response

Reviewer 2

(1) Purpose of study need to clearly explain.

Response1: the purpose of study has been explained.

(2) Literature Review is unnecessarily lengthy.

Response2: Literature review section has been edited.

(3) More information is required about data.

Response3: more information about data have been added.

(4) Conclusion and Future Work, detail about Future Work is missing.

Response4: details about Future Work have been added.

(5) Need to cite the literature about machine learning such as

10.1007/s10118-022-2782-5, 10.1039/D1TA04742F , 10.1002/chem.202103712 , 10.1039/d1ta09762h

Response5: literature about machine learning has been cited references 11 and 12.

Reviewer 3 Report

The Authors attempted to check the performance of several well known classification methods on the set of multivariable data "of the prevalence of diabetes, smoking, obesity, and inactivity in Saudi Arabia from 1999 to 2013". In fact, they repeated (and probably extended) the work which has been done by other researchers, on some other sets of data (see Table 1). In fact, Table 1 only presents the results of previous work, without any deep analysis and discussion of the qualitative features of the classification algorithms applied by those researchers.

The same conclusion can be made when we have a look at the presentation quality of the work presented in the review submission. OK, different algorithms gave different classification accuracy and different inter-class errors, but what significance do these differences have for a researcher who wants to solve a similar problem? Whether the worse accuracy of LD than the accuracy of Gaussian SVM occurs only on this specific data set, or is it a permanent feature of this algorithm, which occurs on data sets representing diabetes? And why the results obtained for groups of men differ from those obtained for women? Etc., etc., etc.

Also, the data in the confusion matrices should be represented as percents, at their current form they are illegible.

Author Response

Reviewer 3

Comments and Suggestions for Authors

-The Authors attempted to check the performance of several well-known classification methods on the set of multivariable data "of the prevalence of diabetes, smoking, obesity, and inactivity in Saudi Arabia from 1999 to 2013". In fact, they repeated (and probably extended) the work which has been done by other researchers, on some other sets of data (see Table 1). In fact, Table 1 only presents the results of previous work, without any deep analysis and discussion of the qualitative features of the classification algorithms applied by those researchers.

Response: In this paper, it has been mentioned that the most of previous studies that use machine learning in diabetes were aimed to diagnosing and detecting diabetes at early stage or modelling the disease progression and complications, however, the aim of our study is studying the trends in the prevalence of diabetes (the epidemiology of the disease) and forecast its future level using risk factors in specific populations such as in KSA. In the literature review section, we review and present some of diabetes studies that applied machine learning techniques.

-The same conclusion can be made when we have a look at the presentation quality of the work presented in the review submission. OK, different algorithms gave different classification accuracy and different inter-class errors, but what significance do these differences have for a researcher who wants to solve a similar problem? Whether the worse accuracy of LD than the accuracy of Gaussian SVM occurs only on this specific data set, or is it a permanent feature of this algorithm, which occurs on data sets representing diabetes? And why the results obtained for groups of men differ from those obtained for women? Etc., etc., etc.

Response: We know that machine learning models may lead to different results (predictions) each time they are trained. Also, they can give different accuracies or level of error after they are evaluated. These differences are caused by differences in training data or differences in platform. Another reason of these differences is the behaviour of these algorithms when dealing with data. The different results obtained for men and women because of the differences in their datasets, for example smoking prevalence is higher among men than women.

-Also, the data in the confusion matrices should be represented as percentages, at their current form they are illegible.

Response: the confusion matrices have been updated.

Reviewer 4 Report

1. The performace measures can be specified in Abstarct

2. The best classification method;s (Weighted KNN) performance can be specified in conclusion also.

3. State the need of all classification algorithm.

Author Response

Reviewer 4:

  1. The performance measures can be specified in Abstract

Response1: performance measure based on accuracy has been specified.

  1. The best classification methods (Weighted KNN) performance can be specified in conclusion also.

Response2: the performance of Weighted KNN has been specified in conclusion.

  1. State the need of all classification algorithm.

Response3: the need of each classification algorithm was included in the classification methodologies section when describing these algorithms.

Round 2

Reviewer 3 Report

I can't see any substantial improvements in the submission, comparing to its previous version. Some pieces of the text added by the Authors do not play any role for increasing the quality of the submission. So, my opinion about the submission must be the same as previously.

There are also some things in the new vesrion, which must be corrected:

  -- in lines 382-383 the Authors say: "... can be found in 382 Entissar et al. [37].", while the position [37] is the work by E. Almutairi, M. Abbod, and T. Itagaki - the first name and the surname of the Author should be used in an unified way;

  -- I do not know why the "Discussion" section has been included into the section describing experimental results;

  -- in some confusion matrices (figures 2 to 8) the sum of non-zero elements is less than 1 (for example in both matrices from fig. 2 that sum is equal to 0.98) - I understand the rules of rounding, but the numbers should be given in an appropriate accuracy.

Of course, it's the Authors' choice how to present the confusion matrices, however I recommend the form of normalized confusion matrices, like in T. Kaichi, T. Maruyama, M. Tada, and H. Saito, "Learning Sensor Interdependencies for IMU-to-Segment Assignment", IEEE Access, vol. 9, 2021, doi: 10.1109/ACCESS.2021.3105801 or Khushaba R. N., Takruri M., Kodagoda S., and Dissanayake G., "Toward Improved Control of Prosthetic Fingers Using Surface Electromyogram (EMG) Signals", Expert Systems with Applications, vol 39, no. 12, pp. 10731–10738, 2012.

Author Response

I can't see any substantial improvements in the submission, comparing to its previous version. Some pieces of the text added by the Authors do not play any role for increasing the quality of the submission. So, my opinion about the submission must be the same as previously.

Response: Regarding the previous points, features and limitations about the machine learning methods have been added to table 1 (only the models that are the same as the proposed once in our current paper).

In the discussion section between line 504 and 515, an additional discussion has been added.

  -In lines 382-383 the Authors say: "... can be found in 382 Entissar et al. [37].", while the position [37] is the work by E. Almutairi, M. Abbod, and T. Itagaki - the first name and the surname of the Author should be used in a unified way.

Response: The name has been corrected.

  - I do not know why the "Discussion" section has been included into the section describing experimental results.

Response: The Discussion section have been updated with more description.

  -In some confusion matrices (figures 2 to 8) the sum of non-zero elements are less than 1 (for example in both matrices from fig. 2 that sum is equal to 0.98) - I understand the rules of rounding, but the numbers should be given in an appropriate accuracy.

Response: the confusion matrices have been updated.

Round 3

Reviewer 3 Report

In fact, the present version does not substantially differ from the previous one. So, I can' change my detailed ratings expressed above, but I recommend to accept the submission.